# The Interplay between α-Synuclein and Microglia in α-Synucleinopathies

**DOI:** 10.3390/ijms24032477

**Published:** 2023-01-27

**Authors:** Jacob S. Deyell, Manjari Sriparna, Mingyao Ying, Xiaobo Mao

**Affiliations:** 1Department of Pathology and Laboratory Medicine, University of California, Irvine, CA 92697, USA; 2UCI Institute for Neurotherapeutics, University of California, Irvine, CA 92697, USA; 3Virginia Commonwealth University School of Medicine, Richmond, VA 23223, USA; 4Hugo W. Moser Research Institute at Kennedy Krieger, Baltimore, MD 21205, USA; 5Department of Neurology, Johns Hopkins University School of Medicine, Baltimore, MD 21205, USA; 6Institute for Cell Engineering, Johns Hopkins University School of Medicine, Baltimore, MD 21205, USA; 7Institute for NanoBioTechnology, Johns Hopkins University, Baltimore, MD 21218, USA; 8Department of Materials Science and Engineering, Johns Hopkins University, Baltimore, MD 21218, USA

**Keywords:** dementia, cognition, microglia, α-synuclein, neuroinflammation, synucleinopathies, neurodegeneration, CNS, immune system, immunology, neuroimmunology

## Abstract

Synucleinopathies are a set of devastating neurodegenerative diseases that share a pathologic accumulation of the protein α-synuclein (α-syn). This accumulation causes neuronal death resulting in irreversible dementia, deteriorating motor symptoms, and devastating cognitive decline. While the etiology of these conditions remains largely unknown, microglia, the resident immune cells of the central nervous system (CNS), have been consistently implicated in the pathogenesis of synucleinopathies. Microglia are generally believed to be neuroprotective in the early stages of α-syn accumulation and contribute to further neurodegeneration in chronic disease states. While the molecular mechanisms by which microglia achieve this role are still being investigated, here we highlight the major findings to date. In this review, we describe how structural varieties of inherently disordered α-syn result in varied microglial receptor-mediated interactions. We also summarize which microglial receptors enable cellular recognition and uptake of α-syn. Lastly, we review the downstream effects of α-syn processing within microglia, including spread to other brain regions resulting in neuroinflammation and neurodegeneration in chronic disease states. Understanding the mechanism of microglial interactions with α-syn is vital to conceptualizing molecular targets for novel therapeutic interventions. In addition, given the significant diversity in the pathophysiology of synucleinopathies, such molecular interactions are vital in gauging all potential pathways of neurodegeneration in the disease state.

## 1. Introduction

Synucleinopathies are a group of heterogeneous diseases that are characterized by the accumulation of misfolded α-synuclein (α-syn) [1,2]. Dementia is commonly seen in synucleinopathies, including Lewy body dementia (LBD), which encompasses dementia with Lewy bodies (DLB) and Parkinson’s disease dementia (PDD) [3,4,5,6,7]. In fact, LBD is the third most common form of dementia in the world, comprising roughly 20–30% of all dementia cases [8]. In addition to LBD, the most common synucleinopathies include Parkinson’s disease (PD) and Multiple System Atrophy (MSA), which can also present with dementia or cognitive deficits [1,2]. Even in Alzheimer’s disease (AD), the most common form of dementia [9], up to 50% of cases reveal co-morbid α-syn pathology [10] with faster cognitive decline and neuropsychiatric dysfunction [6,7]. While the underlying causes of these diseases need to be further elucidated, research has pointed to the potential contributions of aging, environmental factors and genetics [11,12,13].

In synucleinopathies, emerging evidence has shown that misfolded α-syn is a driver in the pathogenesis [14]. It is widely believed that α-syn spread occurs in a prion-like manner, with misfolded α-syn acting as a template for monomeric α-syn [14,15,16]. The spread of misfolded α-syn throughout the brain is further supported by the clinical progression of symptoms. Early in the disease course of synucleinopathies, largely one clinical function is affected, as is supported by models exhibiting α-syn accumulation predominantly in the amygdala and limbic system in DLB [17] or the substantia nigra and basal ganglia in PDD [18,19]. However, these neurodegenerative diseases eventually progress to affect multiple brain areas, including the prefrontal cortex and higher-order cortical areas responsible for executive functions later on in the disease course. Specifically, the disruption of the anterior–posterior circuitry across the medial prefrontal cortex has been implicated in several neurodegenerative diseases [19,20]. This cortical circuitry is highly responsible for behaviors such as planning and anticipation of rewards and threats. These affected brain areas are postulated to contribute to the apathy, blunted response to fear or reward, and general lack of motivation seen in patients with neurodegenerative diseases [21,22,23]. Cell-to-cell transmission of this debilitating protein, which aligns with the clinical course of the disease, is now a generally accepted mechanism underlying neurodegenerative diseases [14]. However, the mechanisms by which α-syn moves from cell to cell are not completely understood. While the neuron-to-neuron transmission of α-syn has been established as an important part of the pathology, there is increasing evidence that microglia play a pivotal role in the spread of α-syn and the overall pathophysiology of synucleinopathies. Current research demonstrates the role of microglia in uptaking, processing, and eventually spreading α-syn. In this review, we aim to delineate the mechanisms by which such processes take place. 

Microglia are the resident immune cells of the central nervous system (CNS), derived from the myeloid lineage [24]. Under physiologic conditions, microglia exist in a homeostatic state, surveilling the brain for any potentially threatening signals, such as pathogens or evidence of neuronal death/neuroinflammation [25]. They regulate neural proliferation, neural differentiation, and some regenerative capabilities of the CNS, making them an ideal target for potential therapeutics [26,27]. In the presence of acute danger signals, microglia activate and take on an amoeboid morphology, leading to the production of pro-inflammatory cytokines through a variety of pathways and aiding in the clearance of foreign antigens [24]. Under pathological states or when the physiological functions of microglia are overwhelmed, they can take on aberrant phenotypes and enable disease progression [28]. Many factors are involved in microglial activation in neurodegenerative diseases, an example being mitochondrial and cellular metabolism dysregulation, which causes downstream inflammation from the build-up of reactive oxygen species (ROS), amino acids, iron, and eventual microglial activation in efforts to clear such cellular degradation products [21,29,30,31].

Microglia have also been repeatedly implicated in the pathogenesis of neurodegenerative diseases such as AD [32,33,34,35], where they internalize and degrade amyloid-β (Aβ) plaques and become pro-inflammatory in nature through the secretion of cytokines and recruit other microglia around the excess extra-cellular protein. Microglial receptors in AD, such as Toll-like receptor 2 (TLR2), recognize α-syn, leading researchers to believe the same mechanism is involved with the progression of PD [36,37]. In fact, in brain PET studies of PD patients, pro-inflammatory microglia can be found dominating the substantia nigra. However, there is a duality to the role of microglia because their actions at the initial stages of synucleinopathies are posited to be more neuroprotective, while in later stages of synucleinopathies, they are postulated as more neurodegenerative [38]. This concept is commonly referred to as the “double-edged sword” of microglial functioning. What causes this shift from neuroprotection to neurodegeneration is yet to be established. Potential theories include temporal influence on microglial functions, as supported by the increased risk of neurodegenerative disease with aging. Over time, it is possible that some combination of aging, environmental factors, and toxins may facilitate the build-up of α-syn and/or increase the propensity of α-syn to misfold, ultimately overwhelming the microglia and reducing its ability to maintain homeostasis. In this review, we aim to expand on such proposed mechanisms regarding microglial uptake, internalization, and processing of α-syn and potential directions for future research.

This review will focus on the following chronological cascade of microglial events in synucleinopathies: 

1. Microglia recognize monomeric, oligomeric, and fibrillar α-syn with cell membrane receptors.

2. Microglia internalize α-syn and take on an “activated state”, which can contribute to a pro-inflammatory environment in the brain.

3. Microglia degrade α-syn while also facilitating the spread of α-syn to other areas of the brain.

## 2. Microglial Interaction with α-Syn Depends on its Structure

### 2.1. Monomeric, Oligomeric and Fibrillar Forms of α-Syn

α-Syn is an intrinsically disordered 140-amino acid protein consisting of an amphipathic N-terminus (residues 1–60), hydrophobic central region (residues 61–95), and acidic C-terminus (residues 86–140) [39]. Synucleinopathies are characterized by the accumulation of α-syn in the CNS. The structural form of the accumulation varies across different patients and different conditions [40,41]. These structural variations can affect the interaction of α-syn with microglia. Structures range from monomeric, oligomeric, fibrillar, and combinational forms, all of which, depending on the receptor interaction, can be internalized in a time-dependent manner [42]. Physiologically, α-syn exists in an unfolded, soluble state but can also exist as a cytosolic α-helix-rich tetramer [43]. Mutations, cellular conditions, and α-syn’s inherent lack of a fixed three-dimensional structure allow it to be aggregated and form random coils, which transition into tertiary helical structures [44]. In particular, the N-terminus has the propensity to form helices. The amphipathic N-terminal region is also known to bind to cellular membranes via its ability to meld with the amphipathic lipid membrane [43,45]. Whether this membrane binding is pathological or physiological depends on several cellular factors. Features such as membrane curvature or lipid rafts/proteins within cell or vesicle membranes increase the propensity of membrane-bound monomeric α-syn to exist as a pair of anti-parallel curved α-helices or a single curved α-helix [45].

In synucleinopathies, α-syn takes on misfolded conformations that switch from α-helix dominance at the N-terminus to being rich in β-sheets, the mechanism by which the familial PD mutation E46K affects the N-terminus [46]. The greater the helicity of the N-terminus, the lower the propensity for aggregation, leading some researchers to believe membrane-bound multimeric α-syn to be protective against synucleinopathies [43,45,46]. However, other findings, such as the disease-associated N-terminal mutation A53T, are known to stabilize the helical sterics of the protein [43]. Such tighter and more stable α-syn becomes more difficult to degrade and thus more likely to aggregate [47]. Aside from the monomeric form, α-syn also misfolds by assembling into oligomeric and multimeric (fibrillar) forms (Figure 1) [48]. These fibrillar forms, when combined with other materials of cellular degradation, can accumulate in the dendritic cell as Lewy bodies. Deposits of Lewy bodies within neurons in the setting of parkinsonism and dementia are pathognomonic for LBD and cause significant downstream neurodegeneration [49]. It is noted that not all fibrillar α-syn is pathogenic, which could be converted by protective mechanisms [50].

In order to determine how the aggregation state of α-syn affects microglial activation, Hoffmann et al., 2016, used BV2 cells, an immortalized murine microglia cell line, and exposed these cells to α-syn monomers, oligomers, and fibrils [42]. This study showed that the fibrillar form of α-syn induced a dose-dependent, highly reactive response from BV2 cells, measuring RNA levels and secreted levels of TNF-α and IL-1β using qPCR and ELISA, respectively. Monomeric α-syn also induced a pro-inflammatory state, but to a lesser degree. Interestingly, oligomeric α-syn did not demonstrate BV2 cell activation. The group also studied the uptake of these three aggregation states of α-syn and demonstrated that only the fibrillar form was taken up by BV2 cells. This interaction between fibrillar α-syn and BV2 cells was further shown to be concentration dependent, with greater concentrations of α-syn leading to increased microglial uptake.

Another group studied the response of primary mouse microglia to α-syn monomers, α-syn oligomers, and α-syn preformed fibrils (PFFs) and utilized cytokine secretion as a measure of the degree of activation of the microglia [55]. Their data in primary microglia support what was previously shown in BV2 cells, as α-syn PFFs induced the greatest inflammatory response in the primary microglia, followed by α-syn monomers, with α-syn oligomers hardly eliciting any response. However, another study demonstrates microglial activation in vivo following hippocampal injection of α-syn oligomers, which could be explained by a few different reasons [56]. The activation of microglia in vivo by oligomeric α-syn could be a secondary response to the activation of other cells, or this finding could simply be due to the fact that microglia in vivo behave differently than in vitro. It should also be noted that oligomeric forms of α-syn are known to be neurotoxic; however, the unstable and heterogenous nature of the oligomeric form often limits its detection [57].

These data, which are summarized in Table 1, demonstrate that all the aggregation states of α-syn have the ability to activate microglia, although oligomeric activation of microglia has not yet been shown to be cell autonomous. More work in vivo should be undertaken to study this oligomeric activation of microglia. These findings should also be further confirmed in a human model for microglia, such as iPSC-derived microglia-like cells (iMGLs). 

### 2.2. Strains of α-Syn in Different Synucleinopathies

Different strains of α-syn exist for different synucleinopathies, and much of the work investigating this field is still ongoing [41,58]. α-Syn strains appear to be specific for a certain synucleinopathy, and they are conserved across patients and have different properties [59]. The misfolding of α-syn was found to have different conformations in MSA and PD. However, little work has been undertaken to specifically show how these different conformations affect microglial target interactions [60]. This does appear to be an emerging field, though, as a recent paper used primary human microglia to show that different strains of α-syn elicit differing strengths of pro-inflammatory microglial responses, but there is no singular portion of the structure in the different fibrils that defines its ability to evoke a neuroinflammatory response [61]. However, some of the folding differences in different fibrils appear to map to the N-terminal region of α-syn, which is the region that is widely believed to bind to cell receptors, so this necessitates follow-up studies [62].

### 2.3. Mutant Forms of α-Syn 

Most cases of synucleinopathies are sporadic, but there are known mutations in the SNCA gene, which encodes α-syn, leading to disease. Major forms of mutant α-syn, including A53T, A30P, and E46K, all have amino acid substitutions in their N-terminal regions [63]. As mentioned before, the N-terminus of α-syn is believed to bind to receptors, so it is interesting to consider whether these mutations may affect their ability to bind to and activate microglia [64]. A group has investigated how microglial activation by mutant α-syn is peptide dependent [65]. The results demonstrated that pro-inflammatory cytokines were most secreted when primary microglia were exposed to the A53T mutant, with less upregulation for the A30P mutant and no increase for the E46K mutant. The E46K mutant was also unable to induce any morphological change in the microglia that it was incubated with, while the A53T and A30P led to altered morphology. These data further support the need to investigate why this differential interaction occurs and the downstream implications. 

## 3. Various Microglial Receptors Interact with α-Syn

### 3.1. Toll-Like Receptors

Toll-Like Receptors (TLRs) are proteins expressed on the cell membrane of microglia and other cells that play a key role in the immune system by recognizing pathogen-associated molecular patterns (PAMPs) [66]. They are most commonly associated with foreign pathogens but have been implicated in pathological protein accumulation as well [67]. Many studies have investigated the involvement of TLRs in microglial recognition of α-syn [67,68,69].

Specifically, microglial TLR4 has been studied with regard to full-length soluble, C-terminally truncated, fibrillar, and A53T mutant α-syn and is important for microglial activation and subsequent uptake of α-syn [70]. TLR4 has been shown to have high expression in patients with synucleinopathies and is crucial for α-syn uptake and downstream inflammatory activity, such as the production of ROS and cytokine release [70]. The removal of TLR4 cascades in vivo through the generation of a TLR4-specific knockout (KO) mouse has been shown to be protective against neuronal death in the striatum and to lead to decreased neuroinflammation [68]. Though this model was not a microglial-specific KO, it demonstrates that TLR4 is intimately involved in PD-related neuroinflammation. 

To better understand the role of TLR4 specifically in microglia, primary microglia from TLR4 KO mice, as well as primary microglia from control mice, were cultured and treated with either full-length soluble, C-terminally truncated or fibrillar α-syn [70]. C-terminally truncated monomeric α-syn is used because it is very prone to form aggregates [71]. Measures of phagocytosis, pro-inflammatory cytokine release, and ROS production were taken for each of these conditions and revealed that TLR4 KO microglia experienced a reduction in all three categories. The primary microglia responded to each of these three forms of α-syn as well, but it was found that the C-terminally truncated monomeric α-syn induced the greatest microglial response. In terms of mutant A53T α-syn, the expression of TLR4 mRNA has been shown to be upregulated in primary microglia after incubation with A53T α-syn. TLR4 has also been shown to be crucial for the clearance of α-syn by autophagy (“synucleinphagy”) [72], which will be discussed later in this review. However, the evidence demonstrating the direct binding of α-syn to TLRs is yet to be published. 

Apart from TLR4, TLR1 and TLR2 have also been shown to functionally interact with α-syn, with TLR2 having a greater described role [73]. It was found that the expression of TLR1 and TLR2 after exposure to mutant A53T α-syn was significantly upregulated in primary microglia [74]. Further studies on TLR2 have demonstrated that α-syn PFFs can activate both BV2 cells and primary microglia through TLR2 [69]. Primary microglia from TLR2 KO mice were also less effective with the uptake of extracellular monomeric α-syn. Additionally, the inhibition of TLR2 in PFF-seeded mice significantly reduced microglial activation in vivo. In humans, post-mortem PD brains demonstrate increased TLR2 expression across both microglia and neurons in comparison to matched controls [74]. It is thought that the monomeric, oligomeric, and fibrillar forms of α-syn interact with TLR2 [56,69]. 

Overall, the TLRs are heavily involved in both monomeric and aggregated α-syn uptake and the activation of microglia, with TLR2 and TLR4 being the major players. The activity of TLRs is implicated in microglial autophagy, increased the release of pro-inflammatory cytokines and increased α-syn clearance. Whether the interaction of microglial TLRs with α-syn protects or progresses the synucleinopathic state is complicated and depends on various factors, including the stage of disease and amount of α-syn build-up. 

### 3.2. Lymphocyte Activation Gene 3

Lymphocyte activation gene 3 (Lag3) is a receptor that is a member of the immunoglobulin superfamily, which binds with pathologic α-syn fibrils [75] and the C-terminus of α-syn [76]. The depletion of Lag3 can reduce the neuronal uptake of α-syn fibrils and the subsequent neuron-to-neuron transmission of pathogenic α-syn. In an in vivo experiment, KO of Lag3 in mice injected with α-syn PFF led to reduced dopaminergic neuron loss and reduced neurodegenerative phenotypes compared to WT mice injected with α-syn PFF. Such findings were also seen when using Lag3 antibodies [75,76]. Similar results of neuroprotection were seen when using murine models overexpressing hA53T α-syn driven by the mouse prion protein promoter [77]. These mice were bred with Lag3 KO mice, and the Lag3 KO mice demonstrated reduced α-syn pathology and microglial activation, along with improvements in behavioral tests. In follow-up studies using Lag3 KO mice and a Lag3 reporter mouse line, Lag3 expression has been proven not only in neurons [76] but also in microglia [78]. In the gene expression profiles of purified microglia isolated at autopsy of individuals without neurodegenerative disease, microglial Lag3 was expressed at levels similar to known microglial marker ITGAM (CD11B), with confirmation at the protein level using DAB staining and immunofluorescence [79]. This is increasingly relevant in the setting of synucleinopathy, where Lag3 levels and microglial activation, in general, are known to be markedly elevated [80]. Functionally, it is interesting to note that microglial Lag3 is being studied as a target for the treatment of depression [78] and that α-syn is more highly expressed in patients with major depressive disorder [81], as many patients with synucleinopathies experience depressive symptoms [82,83,84]. Lag3 could therefore be a possible link between synucleinopathies and depression. Overall, given its expression level in microglia and known functions, Lag3 should be studied as a potentially important microglial receptor for α-syn [75,76,77,78,79,85,86,87].

### 3.3. Triggering Receptor Expressed on Myeloid Cells 2 

Triggering Receptor Expressed on Myeloid Cells 2 (TREM2) is a transmembrane receptor found in several myeloid cells, including microglia, that binds to a host of extracellular ligands, leading to a downstream signaling cascade that promotes survival, proliferation, and inflammation regulation [88]. Some of these ligands that are especially disease relevant include ApoE, Aβ, and the most recently discovered interactor, TDP-43, which is a protein that can become pathologically accumulated in neurodegenerative diseases, such as frontotemporal dementia (FTD) and ALS [89]. Each of these ligands is involved in neurodegenerative disease; however, there remains no documentation as to whether α-syn is a ligand for TREM2. This is an important question to address, as TREM2 mutations have been identified as risk factors for PD [90], and a functional interaction between TREM2 and α-syn has been demonstrated [91]. This functional interaction revealed that TREM2 deficiency leads to increased α-syn induced neurodegeneration and neuroinflammation in vitro and in vivo. 

The importance of interrogating whether a TREM2–α-syn interaction exists is further highlighted by the function of TREM2 mutations in AD. The R47H mutation in TREM2, which has a significant correlation to FTD, AD, and PD, is known to decrease the binding of TREM2 to Aβ, decreasing microglial activation and subsequent clearance of Aβ [92,93]. This finding is further bolstered and shown to be cell autonomous by work conducted using TREM2-KO iPSC-derived microglia monocultures, showing that they exhibit disease phenotypes, including reduced survival, altered phagocytic ability, and impaired chemotaxis [94]. TREM2-deficient microglia become locked in a homeostatic state, indicating the necessity of TREM2 to react to a neurodegeneration-associated stimulus. However, it was also recently published that TREM2 activation over time has the potential to worsen Aβ-induced tau pathology [95], further contributing to the theme of the double-edged sword of microglia and the potential temporal aspect of their function. 

A general consensus across various data implicates TREM2 function as neuroprotective given that mutations in TREM2 confer PD risk. Since TREM2 is known to bind to other pathologically accumulated proteins in neurodegenerative disease, the relationship between TREM2 and α-syn should be further considered.

### 3.4. Metabotropic Glutamate Receptor 5

The group 1 metabotropic glutamate receptor 5 (mGluR5) is a G-protein coupled receptor (GPCR) expressed throughout the brain [96]. Though its expression is highest in neurons, microglia also express mGluR5 [97]. Microglial mGluR5 has been an attractive target due to its involvement in neuroinflammation since the activation of mGluR5 in microglia significantly inhibits their inflammatory response [98]. Researchers built upon this previous knowledge by reporting a pathway that demonstrates mGluR5 involvement in α-syn mediated microglial activation [97]. Using BV2 cells and primary microglia, the group showed that monomeric α-syn physically interacts with mGluR5 and that the activation of mGluR5 dampens microglial activation by monomeric α-syn and protects neurons from toxicity. The group went on to show that the overexpression of monomeric α-syn leads to the degradation of mGluR5 in the lysosome, increasing the potential of α-syn to induce an inflammatory response. The use of a specific mGluR5 agonist prevents the α-syn mGluR5 interaction, thereby preventing the degradation of mGluR5 and presenting a potential therapeutic for synucleinopathies. The group also generated an AAV–α-syn mouse model of PD by using an intrastriatal injection, which is known to form α-syn aggregates. They demonstrated the colocalization and direct interaction of the aggregated form of α-syn with the mGluR5 receptor and demonstrated the capacity of mGluR5 to reduce inflammation in an in vivo model of PD. 

Additionally, aside from α-syn, mGluR5 is also involved in a complex with Aβ with an interestingly sex-dependent interaction, with the cortical and hippocampal mGluR5 binding Aβ in males but not females [99]. A sex-dependent relationship may be an important consideration when investigating the role of mGluR5 with α-syn. Overall, mGluR5 aids microglia in the maintenance of a homeostatic state and decreases neuronal toxicity in the synucleinopathic brain. 

### 3.5. Other α-Syn Receptors

There are several other receptors that have been studied regarding α-syn interaction, and some have been shown to have some relevance for microglia with limited work. The N-methyl-D-aspartate receptor (NMDAR) has previously been identified as a receptor for fibrillar α-syn, and some work has shown that when this interaction occurs on microglia, the NMDA activation of MAPK is blocked [100]. The binding of aggregated α-syn to Cd11b on microglia is important for the activation of NADPH oxidase (NOX2) [101]. The purinergic receptor P2X7 has been shown to interact with both WT and A53T α-syn as well on microglia and activate the p47-PHOX pathway via PI3K/AKT activity, which increases intracellular ROS generation [102]. The FcγRIIB receptor on microglia has been shown to bind to aggregated α-syn, leading to increased SHP-1 activation and the inhibition of phagocytosis [103]. Lastly, the receptor for advanced glycation end products (RAGE) has an alkaline region that was recently found to bind to the acidic C-terminus of α-syn, with preferential binding of α-syn fibrils over other forms [104]. This binding to α-syn fibrils induces neuroinflammation that is reduced in RAGE KO models and with RAGE receptor inhibitors, such as FPS-ZM1. In fact, RAGE, similar to Lag3, is in the immunoglobulin superfamily and physiologically acts as a pattern-recognition receptor on microglia. Further studies should investigate whether there are any other effects of these interactions on microglia and whether receptors such as NMDAR, Cd11b, P2X7, FcγRIIB, or RAGE, could serve as therapeutic targets.

Some receptors have been shown to be important for α-syn interactions in other cell types but have yet to be shown as significant α-syn receptors in microglia. For example, heparan sulfate proteoglycans (HSPGs) are important receptors for the internalization of α-syn by oligodendrocytes, but they have been shown to be far less important for microglia [105]. Neurexins, APLP1, and PrPc have been well studied and proven to interact with α-syn, but they each have very low expression in microglia [86,87]. The a3 subunit of the Na+/K+-ATPase has also not been studied as an α-syn receptor in microglia. The findings involving key microglial receptors for α-syn are summarized in Table 2.

## 4. α-Syn-Induced Inflammation in Microglia

### 4.1. Microglia Uptake of α-Syn

There are a variety of ways in which α-syn can enter microglia. Two prominent modes of entry have been reported to be phagocytosis and macropinocytosis. Several studies have indicated that microglia are able to phagocytose extracellular monomeric and fibrillar α-syn and that TLRs, which are receptors that have previously been described as important in microglial response to α-syn, are important in this phagocytic process [69,106]. Microglia have also been shown to engulf exosomes containing α-syn via macropinocytosis. Specifically, this has been suggested for α-syn-containing exosomes from oligodendrocytes [107]. 

### 4.2. α-Synuclein Induces an Inflammatory Phenotype in Microglia

Microglia are known to exist in a balance between a homeostatic state and a variety of activated states, which are still in the process of being delineated [108]. One such state is disease-associated microglia (DAM), present in diseased brains, such as PD and AD brains [109,110]. Another subset of activated microglia in neurodegenerative disease was termed Microglial Neurogenerative Phenotype (MGnD) [108]. Both DAM and MGnD phenotypes are generally characterized by the downregulation of homeostatic microglial genes and the upregulation of inflammatory genes. 

The homeostatic state presents with more repair and restoration functions, while the DAM/MGnD state is largely responsible for mediating inflammatory responses. The build-up of α-syn elicits a shift in the microglial phenotype from the restorative homeostatic state to the pro-inflammatory state [111]. Neuroinflammation is associated with this microglial phenotypic shift as the activated state is largely responsible for the release of inflammatory cytokines, such as IL-1β, IL-6, IL-8, and TNFα [112]. An additional downstream effect of this inflammatory cytokine production is the activation of the NLR family pyrin-domain-containing 1 (NLRP1) and NLRP3, which contributes to neuronal cytotoxicity [113]. α-syn aggregates trigger the activation of the NLRP3 inflammasome, mostly found in microglia [114]. Microglial NLRP3 inflammasome activation is heightened in the striatum, and, subsequently, the substantia nigra pars compacta (SNpc) in the context of neurodegenerative diseases. α-syn aggregates, through binding to TLR2, cause the transcription factor NF-κB to upregulate NLRP3 expression [114]. Elevated NLRP3 expression leads to additional ROS production, caspase-1 activation, and inflammatory cytokine release, particularly IL-1β. These downstream effects further promote α-syn fibril accumulation and cytotoxicity, particularly of the dopaminergic neurons in the striatum and SNpc. Thus, the microglia enter into a positive feedback loop, with microglial activation further propagating α-syn aggregates [115]. NLRP3 KO mice have reduced motor dysfunction, striatal dopaminergic neuron death, microglial recruitment, IL-1β production, and caspase-1 activation when treated with 1-methyl-4-phenyl-1,2,3,6-tetrahydropyridine (MPTP), a compound widely used to induce PD in murine models [115,116].

The effect of microglial activation by α-syn is further highlighted by the fact that the microglia-specific overexpression of α-syn has the capacity to lead to the severe degeneration of dopaminergic neurons in vivo [111]. Microglia that have accumulated α-syn take on a reactive, pro-inflammatory state and are toxic to their environment. TLR2 and TLR4 KO models have exhibited reduced microglial-mediated inflammation, increased microglial survival, and overall a shift in the microglial surface biomarker profile toward a homeostatic phenotype [117]. Drugs seeking to activate clearance of α-syn by microglia while still maintaining a homeostatic phenotype hold interest as a potential therapeutic. Dendrimer–tesaglitazar, one such drug that acts as a dual PPARα/γ agonist, is currently being explored within cellular and animal models [118].

### 4.3. Involvement of the Adaptive Immune System-MHC Class II, B Cells and T Cells 

Microglia may contribute to inflammation and the pathogenesis of synucleinopathies through their action as antigen-presenting cells (APCs), with the induced expression of major histocompatibility complex II (MHC II) [119]. It has been described that upon injection of α-syn PFFs in the striatum of rats, microglia acquire early and sustain MHC II expression [119]. Microglia are not the only cells that begin to overexpress MHC II; the group detected that both Iba1+ and Iba1- cells expressed MHC II, where Iba1 is a microglia-specific marker in the CNS. Another group also described a similar phenomenon of microglial MHC II expression upon exposure to α-syn and reported that this process occurs prior to the onset of neurodegeneration [120]. Additionally, genome-wide association studies have implicated haplotypes of MHC II with PD risk [121]. Certain forms of MHC II or the degree of its expression may be detrimental as microglia can present α-syn on MHC II, leading to CD4+ T cell activation, proliferation, and downstream pro-inflammatory cytokine release that mediates further neuronal degradation. In fact, CD4+ T cells have been shown to gather around α-syn aggregates accompanied by elevated chemokine receptor (CXCR4 and CXCL12) expression in the CSF of LBD patients, further implicating the chemokine-directed movement of CD4+ T cells to the site of abnormal α-syn [122,123]. This evidence suggests that MHC II expression may be a physiologic response of microglia to an unrecognized antigen that ends up furthering inflammation and neurodegeneration in synucleinopathies. 

There is also evidence of the downstream activation of B cells through interactions with CD4+ T cells in the CNS. PD patients with progressing symptoms are known to have higher levels of high-affinity α-syn antibodies compared to patients that stabilize in a prodromal PD state [124], thereby supporting antibody production by B cells due to activation by CD4+ T cells. In vitro models have proven that soluble antibody-α-syn complexes further neurotoxicity by activating NLRP3 inflammasomes in microglia [68], thereby implicating antibodies and, in turn, B cell involvement within neuroinflammatory cascades in PD. 

There is some evidence that the microglial processing of α-syn facilitates CD8+ T cell activity to a greater extent than CD4+ T cell activity [125]. In post-mortem brain analyses, a higher density of CD8+ T cells, as opposed to CD4+ T cells, was noted in both the brain parenchyma and brain perivascular spaces. Regardless of which type of T cell is present in the diseased brain, the presence of such T cells heavily implicates the role of microglial cells as the APC that activates CD4+ cells. In the event that this microglial presentation to CD4+ cells causes Th1 differentiation, the Th1 CD4+ T cell can release IL-2 and activate a potent CD8+ T cell response. Activated CD8+ cytotoxic T cells in the setting of extracellular α-syn can also occur with the neuronal presentation of α-syn on MHC class I, but whether microglial MHC II or neuronal MHC I presentation of α-syn is predominant remains unknown.

Although the above sections point to the fact that the microglial-dependent activation of T cells is detrimental to the CNS and contributes to the pathogenesis of the disease, it is important to note that T cells may also play a positive role by modulating reducing pathological α-syn in the brain [126]. Immunocompromised mice that lack T cells, B cells, and Natural Killer (NK) cells received intrastriatal injections of α-syn PFF along with reconstitution of T cells, B cells, NK cells, or no cells reconstituted. When compared to WT mice, the immunocompromised mice demonstrated an eight-fold increase in substantia nigra pathology of phosphorylated α-syn. However, by reintroducing T cells into the mice, the phosphorylated α-syn pathology was significantly decreased. 

Distinctive epitope regions of α-syn are implicated in neurodegeneration. Notably, epitopes derived from the Y39 region are known to be displayed by MHC II beta chains with the DRB5*01:01 and DRB1*15:01 alleles in a majority of PD patients [127]. Another antigenic region on α-syn, namely S129, drives immune responses in patients without the HLA alleles typically associated with neurodegeneration, such as the DRB alleles that recognize the Y39 region. In fact, such a large majority of α-syn is phosphorylated at S129 in synucleinopathies, wherein antibodies against phosphorylated α-syn are used as a marker of pathology [128]. Interestingly, a recent study used pentameric formyl thiophene acetic acid (pFTAA), a compound with conformation-dependent spectral properties in the setting of amyloidosis, to examine α-syn inclusions in four different transgenic α-syn overexpression mouse models that differed based on: the promoter-driving expression of α-syn, the presence of a mutation in the α-syn being expressed, and whether human or mouse α-syn was overexpressed [129]. Differences could not be determined in the distribution and appearance of the α-syn lesions, even though all four mouse lines have very different disease progressions. However, what the study did find using pFTAA, is that microglia had α-syn aggregates, which were further confirmed using antibodies for the N-terminus and NAC domain of α-syn. These aggregates could not be detected using antibodies specific to the C-terminus of α-syn, including an antibody for α-syn phosphorylated at S129, indicating that microglial aggregates lack this common phosphorylation site and have a distinct form from neuronal aggregates. The lack of this phosphorylation site also explains a potential reason why microglial aggregates of α-syn have been overlooked. While the authors acknowledge that microglial aggregates could be an artifact of murine α-syn overexpression systems, it is important to follow up on this finding. 

## 5. Processing and Spread of α-Syn

### 5.1. Degradation of α-Syn Post-Microglial Uptake Linked to Microglial Autophagy

Ingested α-syn is degraded inside microglia via a TLR4-dependent mechanism [72]. TLR4 activation results in the subsequent activation of the NF-κB cascade and transcription of p62, an autophagy receptor necessary for forming α-syn–ubiquitin complexes for subsequent autophagy in microglia. The autophagy of extracellular α-syn has been termed “synucleinphagy”. In the synucleinopathic CNS, p62 recognizes and binds internalized and ubiquitinated α-syn within microglia to bring it under proteasomal degradation. High levels of proteasomal degradation can cause the microglial cell to undergo autophagy. Interestingly, the KO of microglial autophagy genes such as Atg7, a gene encoding a ubiquitin-conjugating enzyme, and Atg14, a gene encoding PI3-kinase VPS34, prevent the degradation of α-syn within the microglial cell through unelucidated mechanisms, resulting in increased dopaminergic neuronal death. Overall, microglia can clear extracellular α-syn through autophagy, and a decrease in autophagy flux can promote neurodegeneration.

### 5.2. Secreted Proteins for α-Syn Degradation

Although minimal, research on proteins secreted by microglia demonstrates their function in processing extracellular α-syn. Two of these enzymes include insulin-degrading enzyme (Ide) and matrix metalloproteases (MMPs), including MMP1, 3, 9, and 13 [130,131]. Ide has been found to act in two different manners [131]. For one, it inhibits the formation of α-syn fibrils via binding to oligomeric α-syn and preventing further aggregation. Additionally, it acts as a protease to degrade α-syn. MMPs are endopeptidases that are released by neurons and glia in response to inflammatory stimuli and share similar proteolytic activity to Ide. These proteins that degrade α-syn are an attractive therapeutic target, and a recent group showed that the previously mentioned dendrimer–tesaglitazar increased the expression of Ide and MMP9 [118]. However, MMPs simultaneously have the propensity to amplify an immune response. It was shown that microglia exposed to recombinant MMP13 take on a pro-inflammatory phenotype, adopting an amoeboid morphology and releasing pro-inflammatory cytokines [73], again alluding to the double-edged sword presented by microglia.

We will now shift our focus to the spread of α-syn, which is still an area of study that is still very much emerging. There appear to be several mechanisms of spread, some of which are established and likely some that are still not known. Here, we will focus on the mechanisms specific to microglia. We feel microglia are relevant, especially because of the recent finding that microglia assist in the propagation of Aꞵ throughout the brain [33]. 

### 5.3. Exosomal Transmission of α-Syn

A poorly understood mechanism in synucleinopathies is the spreading of α-syn throughout the CNS. In recent years, exosomes have become of particular interest in many diseases, including synucleinopathies. It has been shown that neuronal cells overexpressing α-syn can release exosomes containing α-syn to normal, healthy neuronal cells and promote the clustering of SNARE complexes at neuronal presynaptic terminals [132,133,134,135]. Fluorescence resonance energy transfer (FRET) studies demonstrate a folding pathway for α-syn where the prion-like protein transitions from its monomeric, natively unfolded cytosolic form to a physiologically functional, multimeric membrane-bound form. The multimeric membrane-bound form or even excessive membrane-bound monomers interact with SNARE proteins through their more helical structure and are shown to disrupt the fusion of dopamine-containing exosomes, especially in the early stages of the disease process [136]. This disruption has been proven to show later synapse instability, especially in dopaminergic neurons. α-syn’s disruption of dopamine-carrying exosomes from docking and fusing begs the question of whether α-syn interacts further with exosomes. Can it bind to exosomal membranes via the same mechanism as it binds to the lipids in neuronal membranes? Given its role in disrupting dopamine-containing exosomes specifically, can α-syn spread to less overwhelmed dopaminergic neurons through the exosome that was unable to fuse? Given that Aβ (in Alzheimer’s disease), tau (in numerous neurodegenerative diseases), prions (in transmissible spongiform encephalopathies), α-syn (in synucleinopathies, including PD [137]), and superoxide dismutase 1 (in ALS) have been previously proven to spread via exosomes [138], experiments answering such questions may provide important insight behind the transition from physiologic α-syn to pathologic α-syn function. In fact, neuronally derived exosomes can be found even in the peripheral blood and are being explored as a blood-based biomarker to diagnose synucleinopathy [139,140].

It was recently revealed that in addition to neurons, microglia are also key players in α-syn exosome production [141,142]. Through in vitro experiments studying primary murine microglia, a group found that microglia can secrete α-syn exosomes capable of inducing further aggregation in the neurons that received them. This aggregation was increased when pro-inflammatory cytokines produced by microglia were present. By using an exosome formation inhibitor in microglia, the group noted reduced α-syn transmission. Similar effects were seen in vivo [143]. The group proposed a disruption in autophagy flux as a potential mechanism for microglial exosome release, as the connection between autophagy inhibition and exosome formation has been previously described [144]. Microglia that were treated with α-syn preformed fibrils were found to have increased levels of PELI1, which is an E3 ubiquitin ligase. PELI1 led to the degradation of lysosomes and the inhibition of autophagy. This inhibition of autophagy might promote the transfer of α-syn to other cells via exosomes. Another group noted similar involvement of microglia with α-syn exosomes [145]. They found that when exogenous exosomes from the plasma of PD patients were injected into the brains of mice, microglia had a great tendency to take up these exosomes and become activated. They reported a similar mechanism of microglial autophagy dysregulation, leading to increased intracellular α-syn accumulation and the subsequent secretion of α-syn. 

### 5.4. α-Syn Transmission via Nanotubes

Recent investigations have shown that under a high α-syn burden, microglia form a functional network that permits splitting the burden of α-syn accumulation with other microglia [146]. The intercellular exchange of fibrillar α-syn had been previously demonstrated in other cells, such as neurons and astrocytes, but only recently in microglia. This exchange is believed to take place both through gap junctions and tunneling nanotubes. At baseline, microglia do have intercellular connections, but it was found that the presence of aggregated α-syn increases the number of said nanotubes. Healthy microglia are also able to donate mitochondria to microglia that are overloaded with α-syn in order to assist with degradation and minimize the formation of ROS. Notably, this method of α-syn “sharing” was impaired in microglia that harbor the PD-relevant LRRK2 G2019S mutation.

With this exchange of α-syn likely comes a benefit for the donor microglia but also a disadvantage for the acceptor microglia. The donor microglia become more capable of dealing with a reduced fibrillar α-syn burden; however, now a new microglia cell has been “infected” with α-syn, and given the prion-like nature of α-syn, this can lead to the seeding of more α-syn, causing an even greater burden. However, the donor cell also transfers mitochondria, which aid the acceptor cell in mitigating its newfound α-syn burden. Ultimately, the network of two microglia processing α-syn reduces the number of inflammatory cytokines and ROS formed. This method of α-syn transfer between microglia remains to be fully demonstrated in vivo. 

## 6. Discussion

It is evident that the role of microglia in synucleinopathies is complicated and vast. The culmination of the data above suggests that with the accumulation of α-syn in the brain, different forms of α-syn can bind a slew of receptors on the microglial surface. The binding of α-syn to microglia allows for the detection and uptake of α-syn by the microglial cell. This can lead to the activation of a pro-inflammatory state in microglia. The accumulation of α-syn, interactions with the receptor, and the spread of α-syn are summarized in Figure 2.

Depending on the receptors activated by α-syn (i.e., TLRs or mGluR5), microglia can carry out different downstream effects, such as releasing inflammatory cytokines, activating the adaptive immune system, activating proteasomal degradation of α-syn, activating the synucleinphagy pathway, or spreading α-syn to other brain cells via exosomes or nanotubes. Under non-pathological conditions or early disease states, the α-syn would ideally be cleared from the brain, and neuroinflammation would subside. Under pathological conditions, excess α-syn build-up results in the formation of α-syn fibrils. Increased α-syn in the brain promotes TLR expression, which results in excessive neuroinflammation, perpetuating this cycle. Excess α-syn in the brain also promotes neuronal and microglial uptake of α-syn [76,106], causes microglial activation [106,148], creates oxidative stress [149], and leads to the conversion of reactive astrocytes [148], ultimately causing neuronal death [150]. Emerging studies have shown that ROS scavengers [151,152] and microglial inhibition [148] can significantly protect neurons from pathogenic α-syn.

The evidence described in this review highlights the fact that microglia have the potential to play both a neuroprotective and neurodegenerative role in synucleinopathies. While they can aid in the clearance of the α-syn burden, they can also propagate the α-syn burden and create an inflammatory environment. This makes therapeutic design difficult. Therapeutic efforts should be directed toward developing molecules that can activate microglial phagocytosis of α-syn and subsequent degradation of the protein while minimizing associated detrimental aspects of neuroinflammation. The nanobody that specifically recognizes pathogenic α-syn fibrils might help the clearance in microglia and other cells [153]. Ideally, such interventions would promote more of an overall homeostatic microglial phenotype.

However, many questions remain unanswered, such as the mechanisms behind fibrillar shifts in α-syn structure and how exactly this affects microglial binding in vivo. Furthermore, upon microglial uptake, it remains unclear what causes microglia to favor spread and neuroinflammation over contained degradation. Additionally, the pathophysiology of any associated neuroinflammation is regulated by many neural circumstances. It is unclear to what extent microglial activity contributes to the onset versus the progression of synucleinopathies. Some studies suggest that systemic inflammation precedes motor and cerebral decline. Symptoms from general inflammation such as constipation from intestinal inflammation associated with certain gut flora and appetite changes due to neuroinflammation near the hypothalamus, can precede cognitive or motor decline by decades [154,155,156,157]. Microglial inflammation is also posited to occur concurrently with or after such systemic symptoms appear, typically very early in the disease process [154,158]. One PET study in PD patients used a ligand that specifically binds to activated microglia and visualized significant signals even in early disease states [159]. According to some studies, an estimated 30% of total dopaminergic neurons are already lost by the time a synucleinopathy is diagnosed based on late-disease cognitive and motor symptoms [156], thus implicating the role of microglia in facilitating such neuronal death early in the disease course. Future experiments should delineate whether microglia are actively involved in the propagation of α-syn throughout the brain, as it is quite plausible that the same could be true for what was recently shown for Aβ.

The information we have described in the review highlights important areas of investigation for the interactions between microglia and α-syn. Some of these areas are more developed (e.g., interaction with TLRs), while others remain understudied and present a great opportunity for increasing understanding (e.g., TREM2). We have synthesized information based on the current knowledge of receptors and downstream effects and present them in a consolidated format. Future research will be important in discerning the most important microglial receptors for α-syn that should be targeted. We believe the studies presented show that microglia are key players in synucleinopathies and that this review guides researchers to note the current gaps in knowledge so that they can be addressed. This will aid in the overall goal of understanding all aspects of the interaction between microglia and α-syn and if therapeutics could be developed to target microglia to aid in the treatment of synucleinopathies, increasing the neuroprotective effects of microglia while limiting their detrimental ones. Some examples, based on the topics described in this review, include antibodies targeting receptors to block the microglia–α-syn interaction or drugs to aid with microglial autophagy. The true purposes, both beneficial and detrimental, of neuroinflammation are not yet completely defined. The challenge in the field of microglia and neurodegenerative disease in the past was the lack of robust models of microglia; however, this is changing with improved stem cell technologies [160,161,162]. Microglia are a key component to understanding the pathogenesis of synucleinopathies and neurodegenerative diseases in general, so following up on the gaps in the research presented in this article will aid in developing effective disease-modifying therapies. In the next section, we will provide some examples of steps to be taken. 

## 7. Conclusions, Limitations, and Future Directions

Altogether, microglia appear to be critical to the pathological progression of synucleinopathies. The studies presented in this review represent many different interactions and pathways, but there remains a lot unknown about the specifics of microglial states and what role they play in synucleinopathies. The limitations of the current literature likely stem from the fact that, for many years, robust models of microglia were not available for study. BV2 cells do not recapitulate all aspects of human microglia, and primary murine microglia are difficult to maintain in a homeostatic state in culture. The advent of iMGLs revolutionized the field and has allowed for more rapid and physiological discovery [94,160,161,162]. 

Future work should be conducted to better identify the molecular mechanisms of microglia that are at play in synucleinopathies. One potential route of investigation could be to conduct CRISPRi screens in iPSC-derived microglia that are exposed to different forms of α-syn to assess what molecules are, in fact, critical for the downstream effects seen. This could be achieved using the recently established platform of iTF-Microglia with CRISPRi/a technology [161]. 

Because pathogenic α-syn can enter microglia and cause microglial activation, which can subsequently induce neurotoxic reactive astrocytes [148,163], it is best to develop a specific agent (e.g., nanobody [153]) that targets the intracellular pathogenic α-syn to reduce the consequential chain reaction. Oxidative stress is another hallmark of α-synucleinopathies that has a strong interplay with neuroinflammation [164], so developing agents to reduce α-syn-induced oxidative stress (e.g., nanomaterials and natural molecules [50,151,152]) could be an effective strategy against further neuroinflammation and neurodegeneration.

Additionally, in order to gain a better understanding of these unanswered questions, more robust model systems should be used. iMGLs derived from patients with synucleinopathies will be key to understanding the role that microglia play in these diseases. iMGLs have been shown to be robust models of microglia in vitro and can recapitulate some disease phenotypes [94,160]. Analyzing iMGL interactions with α-syn alone and in a triculture system with astrocytes and neurons will help to elucidate relevant disease pathways and can allow for the screening of potential therapeutics.

## 8. Methods

We performed a systematic review of the literature to investigate how microglia interface with α-syn. All searches were conducted using PubMed. For the section on microglial receptors for α-syn, we performed the search using the following keywords: (“microglia”) AND (“synuclein”) AND (“receptor”), which yielded 201 results. From this point, we reviewed the titles and abstracts of all papers, noting that several papers focused on certain receptors for which specific searchers were subsequently carried out. Specific searches were also conducted for the literature surrounding microglia and known receptors for α-syn that have been demonstrated as key for extracellular interactions [165], as well as for microglial receptors that have been demonstrated as relevant to neurodegenerative disease (e.g., TREM2 [91]) to ensure our review was complete and comprehensive. We reviewed the title and abstract of all hits to assess relevance, and papers were included if there was substantial evidence suggesting the importance of receptor interactions in the pathophysiology of synucleinopathies. Similar searches were conducted for the subsequent sections replacing the search term (“receptor”) with (“uptake”) = 39 results, (“phagocytosis”) = 51 results, (“clear”) = 50 results, and (“spread”) = 84 results, with subsequent specific searches after finding hits that were relevant to the scope of the paper and had substantial evidence. Titles and abstracts of all hits were read and considered within the scope of our review, with hits being included if there was substantial, cutting-edge, or promising evidence for involvement with microglia and synucleinopathies. 

## Figures and Tables

**Figure 1 ijms-24-02477-f001:**
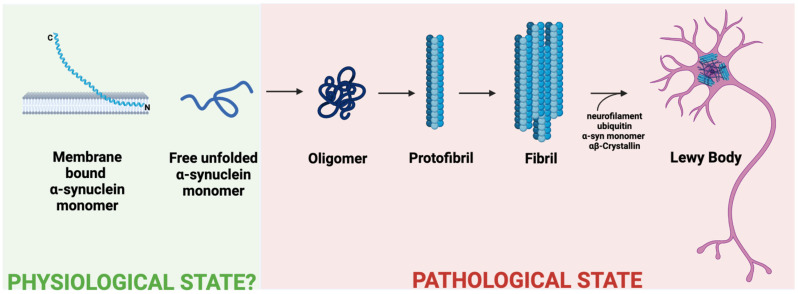
Progression of α-syn from potentially physiological to pathological states. Various misfolded forms of α-syn exist in the pathological state, and whether membrane-bound α-syn is physiological remains controversial [51,52]. Membrane features influence membrane-bound forms of α-syn, allowing it to exist as a pair of anti-parallel curved α-helices or a single curved α-helix (as shown) [53]. Various forms of monomeric and fibrillar α-syn combine with neurofilaments, ubiquitin, and αβ-Crystallin to accumulate in neurons as Lewy bodies [54].

**Figure 2 ijms-24-02477-f002:**
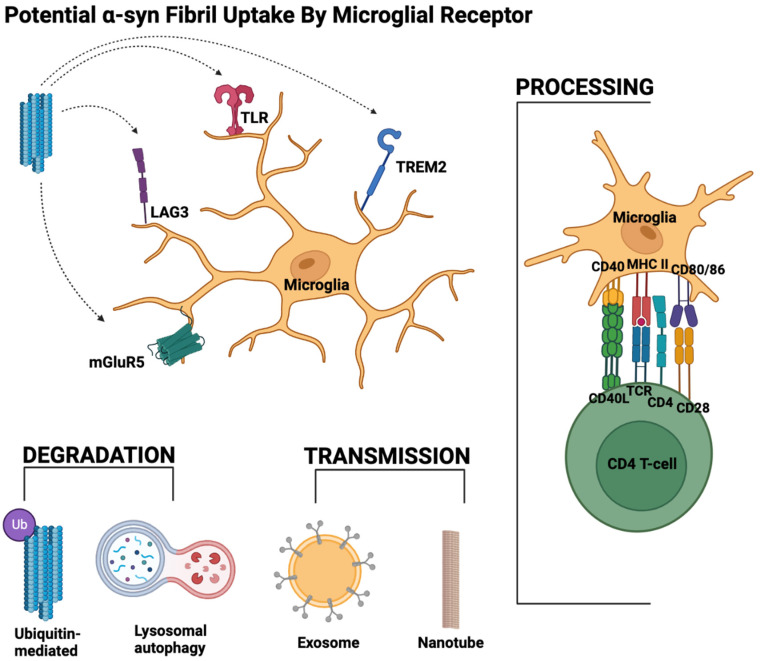
Microglial interaction with α-syn and downstream effects. Summary of the pathologic aggregation of α-syn in the CNS, the potential receptor interactions on the microglial surface, and the downstream effects after uptake of α-syn, including degradation, transmission, or processing. α-syn uptake potentially occurs via interacting with TLRs, LAG3, TREM2, and/or mGluR5 although repeated evidence of such interactions is pending. After uptake, α-syn can be degraded by the microglial cell through ubiquitin-mediated proteasome activation or lysosomal-mediated autophagy. Instead of uptake, α-syn can also be transmitted to other microglia, particularly if the existing microglia have a high α-syn burden. Transmission of α-syn occurs through exosomes or via nanotubes connecting microglia. Lastly, a microglial cell can process and present α-syn on its MHCII receptor to activate downstream immune cascades [147].

**Table 1 ijms-24-02477-t001:** Interaction of microglia with different aggregation states of α-syn. A summary of the ability of different aggregation states of α-syn to activate microglia as seen in the BV2 cell line, murine primary microglia and in vivo in a mouse brain, along with the method of detection used. The strength of the activation is shown for in vitro work by strongest (++), strong (+), or absent (-), whereas in vivo work is marked as present (X) or unknown (?).

	BV2 Cells	Primary Microglia	Mouse (in vivo)
Method of Detection	qPCR (from BV2 RNA) and ELISA (from cell culture supernatant) for inflammatory mediator production (TNFα and IL-1β)	ELISA (from cell culture supernatant) for inflammatory cytokines (TNFα and IL-1β)	Various methods, including analysis of microglial morphology with Iba1 staining and IHC for IL-1β
Monomer	+	+	?
Oligomer	-	-	X
Fibrillar	++	++	X

**Table 2 ijms-24-02477-t002:** Summary of important α-syn receptors in microglia for future investigation. Compilation of the receptors described in this review, the form of α-syn they bind to, and any known downstream effects.

Receptor	Known Interactor?	Aggregation State of α-Syn Known to Interact with Receptor	Downstream Effect
TLR	Yes	Monomeric and aggregated forms	Phagocytosis of α-syn Secretion of ROS and pro-inflammatory cytokines Synucleinphagy
Lag3	Yes, but not yet involving microglia	Aggregated forms	Unknown
TREM2	Unknown	Not yet known	Unknown, but likely survival, phagocytosis and proliferation
mGluR5	Yes	Monomeric and aggregated forms	Neuroprotection Dampens immune response
NMDAR	Yes	Aggregated forms	Decreased homeostatic microglial activity
Cd11b	Yes	Aggregated forms	Increased microglial oxidative stress
P2X7	Yes	Aggregated forms	Increased microglial oxidative stress
FcγRIIB	Yes	Aggregated forms	Inhibition of phagocytosis
RAGE	Yes	Monomeric and preferential binding of aggregated forms	Neuroinflammation evidenced by secretion of TNF-α, IL-1β, and IL-6

## Data Availability

Not applicable.

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
