# Peer review of "The Interplay between α-Synuclein and Microglia in α-Synucleinopathies"

_ijms, 2023, doi:10.3390/ijms24032477_

Round 1

Reviewer 1 Report

Review of a manuscript “Molecular Mechanisms of α-synucleinopathies: The Interplay Between α-synuclein and Microglia” by Jacob Sahag Deyell and coauthors submitted to IJMS.

A group of neurodegenerative diseases called synucleinopathies are serious disorders for which there is no effective treatment modifying the course of these illnesses and no reliable diagnostic biomarkers for early identification of their beginning. Thus, research is needed to better understand molecular and cellular mechanisms of their pathogenesis. The author of this review  combined and summarized published data about the role of α-synuclein in interaction with microglia. This is an important area of biomedical research, and the review will be interesting for the readership of the journal. The following corrections and additions should be made.

Lines 83-85: After the sentence ”In synucleinopathies, α-syn takes on misfolded conformations that switch from α-helix dominance to being rich in b-sheets. a-syn also misfolds by assembling into oligomeric and multimeric (fibrillar) forms (Figure 1)” the authors should add a citation on a recent review of three types of synuclein : ”Synucleins: New Data on Misfolding, Aggregation and Role in Diseases. Biomedicines. 2022 Dec 13;10(12):3241. doi: 10.3390/biomedicines10123241.

Line 92: ”..a group used BV2 cells” This sounds weird and unclear. Better “Hoffmann et al., 2016 used.”

Line 94”… measured by qPCR” Unclear. What exactly was measured

Line 138 2.3 Mutant forms of α-syn

“A53T, A30P, and E46K, all have mutations on their N-terminal regions [21]”. Should be corrected as follows:”A53T, A30P, and E46K, all have amino acid substitutions on their N-terminal regions [21].

Line 291 “4.α-. syn-Induced Inflammation in Microglia” should be corrected as “4. α-Syn-Induced Inflammation in Microglia”.

Line 301 “4.2α-. Synuclein Induces an Inflammatory Phenotype in Microglia” should be corrected as“4. 2 α- Synuclein Induces an Inflammatory Phenotype in Microglia”

Line 281 : ”Another antigenic region on α-syn, namely S129, drives immune response in patients…” It would be interesting to discuss here the effect of α-synuclein phosphorylation at serine 129 and the results published by Tanriöver et al in: “Prominent microglial inclusions in transgenic mouse models of α-synucleinopathy that are distinct from neuronal lesions. Acta Neuropathol Commun. 2020 Aug 12;8(1):133. doi: 10.1186/s40478-020-00993-8.

Line 443. “5.4(-. Syn Transmission via Nanotubes” should be replaced on “5.4  α - Syn Transmission via Nanotubes”

Lines 504-505:”Some studies suggest that inflammation precedes symptoms of motor and cerebral decline, while others suggest they are concurrent with symptom progression.” The authprs should supply references for both examples.

Lines 513-514: ”…how they play into synucleinopathies” should be rewritten as “…and what role they play in synucleinopathies.”

Reviewer 2 Report

5 January 2023 

Review on the manuscript titled ‘Molecular Mechanisms of α-synucleinopathies: The Interplay Between α-synuclein and Microglia’ by Deyell JS et al, submitted to International Journal of Molecular Sciences (IJMS)

Manuscript ID: ijms-2161520 

Dear Authors, 

Deyell and colleagues in the present article entitled ‘Molecular Mechanisms of α-synucleinopathies: The Interplay Between α-synuclein and Microglia’, investigated the current status of knowledge of microglia role the pathological progression of synucleinopathies. For this purpose, they selected some relevant studies that focused on chronological cascade of microglial events in synucleinopathies, like how microglia recognize monomeric, oligomeric, and fibrillar alpha-synuclein (α-syn) with cell membrane receptors; how microglia internalize α-syn and take on an “activated state”; how microglia degrade α-syn, while also facilitating the spread of α-syn to other areas of the brain.

The main strength of this paper is that it addresses an interesting and timely question, describing the potential downstream effects of the α-syn interaction with microglia, focusing on how microglia process α-syn and how microglia can aid in the spread of α-syn to other regions of the brain. In general, I think the idea of this article is really interesting and the authors’ fascinating observations on this timely topic may be of interest to the readers of IJMS. However, some comments, as well as some crucial evidence that should be included to support the author’s argumentation, needed to be addressed to improve the quality of the manuscript, its adequacy, and its readability prior to the publication in the present form, in particular reshaping parts of the Introduction and Discussion sections by adding more evidence and theoretical constructs.

Please consider the following comments:

1.      Title: Please present the title concise, self-explanatory, and stating the most important message of this review article.

2.      Abstract: Please abridge the abstract to 250 words, proportionally presenting the background, the objectives, the short summary, and the conclusion. The background should include the general background (one to two sentences), the specific background (two to three sentences), and current issue addressed to this review (one sentence). The end of the short summary should include one to two sentences which put the result into a more general context. The conclusion should include one sentence describing the main message using such words like “Here we highlight”, the potential and the advance this article has provided in the field, and finally a broader perspective (two to three sentences) readily comprehensible to a scientist in any discipline.

3.      A graphical abstract is highly recommended.

4.      Keywords: I would suggest adding ‘synucleinopathies’ as a keyword. Please list ten keywords and use as many as possible in the title and in the first two sentences of the abstract.

5.      A graphical abstract that will visually summarize the main findings of the manuscript is highly recommended.

6.      I would ask the authors to clarify the criteria they decided to use for studies’ collection in their review: they should specify the requirements used to decide whether a study met the inclusion/exclusion criteria of the review, describe whether they included a balanced coverage of all information that is actually available, whether they have included the most recent and relevant studies and enough material to show the development and limitations in this field of interest. Finally, I believe that they should briefly present results of all statistical syntheses conducted.

7.      The objectives of this study are generally clear and to the point; however, I believe that there are some ambiguous points that require clarification or refining. In my opinion, the authors should be explicit regarding how they sought to assess how the aggregation state of α-syn affects microglial activation, since this is the key aim of this review.

8.      Introduction: This section is well-written and nicely presented, with a good balance of information about the binding α-syn to microglia that leads to activation of a pro-inflammatory state in microglia. Nevertheless, I recommend that the authors fully expand this section by starting with the general background, providing with information on the main constructs of this review, which should be acknowledged to a reader in any discipline, and with the implication of this review to the general background, and make persuasive enough to put forward the main purpose of current research the authors conduct and the specific purpose the authors have intended by this review. I would like to encourage the author to fully extend the introduction starting with the general background, proceeding to the specific background, and finally the current issue addressed to this study, leading to the objectives which is missing in this section. Those main structures should be reorganized in a logical and cohesive manner. I believe that more information on links between α-synuclein aggregation and the effect of inflammation in neurodegeneration, will provide a better and more accurate background. Thus, I suggest making such effort to provide a brief overview of the pertinent published literature on development of α-syn pathology, for example focusing on ‘Mitochondrial impairment: a common motif in neuropsychiatric presentation, and the link to the tryptophan–kynurenine metabolic system’ and on how activation of microglia is one of the hallmarks of the neuroinflammatory promoting neurodegeneration (https://doi.org/10.3390/ijms23136991; https://doi.org/10.3390/ijms23136991; https://doi.org/10.31083/j.fbl2709265). I believe that adding information from these studies may improve the theoretical background of the present article and its argumentation by highlighting how α-syn pathology in the brain correlates with progression of neuropathological changes in specific brain areas, like prefrontal cortex and how ‘The human ventromedial prefrontal cortex support fear learning, fear extinction or both’.

9.      Synuclein Induces an Inflammatory Phenotype in Microglia: In this section, authors mentioned how neuroinflammation is associated with the release of inflammatory cytokines. In this regard, I believe that it could be useful to the authors to deepen information about how the release of α-syn in early disease possibly via the activation of the inflammasome NLRP3 pathway, stimulates microglia to release proinflammatory cytokines, which are detrimental to dopaminergic neurons.

10.  Discussion: After the short summary of this review detailed in the previous sections, I recommend that the authors discuss and fully develop this section by focusing on the current issues addressed to this review. I suggest, toward the end of this section, clearly stating the potential of this article complementing as the extension of the previous understanding, the implication of the authors’ opinion, how this article could facilitate future research, the ultimate goal, the challenge, the knowledge and the technology necessary to achieve this goal, the statement about this field in general, and finally the importance of this line of research.

11.  I would ask the authors to include a proper and defined ‘Limitations and future directions’ section before the end of the manuscript, in which authors can describe in detail and report all the technical issues that could be brought to the surface.

12.  References: Authors should consider revising the bibliography, as there are several incorrect citations. Indeed, according to the Journal’s guidelines, they should provide the abbreviated journal name in italics, the year of publication in bold, the volume number in italics for all the references. Please cite more references. Typically, a review article like this cites more than 150 references.

Overall, the manuscript contains 2 figures, 2 tables and 92 references. I believe that the manuscript may carry important value in describing the potential downstream effects of the α-syn interaction with microglia, focusing on how microglia process α-syn and how microglia can aid in the spread of α-syn to other regions of the brain. I hope that, after these careful revisions, the manuscript can meet the Journal’s high standards for publication. I am available for a new round of revision of this article.

Best regards, 

Reviewer

Round 2

Reviewer 2 Report

20 January 2023 

The 2nd review on the manuscript titled ‘Molecular Mechanisms of α-synucleinopathies: The Interplay Between α-synuclein and Microglia’ by Deyell JS et al, submitted to International Journal of Molecular Sciences (IJMS)

Manuscript ID: ijms-2161520 

Dear Authors, 

I am pleased to see that the authors took my comments seriously and have solved most of issues I raised in the previous round of the peer-review session. Currently, the manuscript is a well written and nicely presented review paper discussing the current status of knowledge of microglia role the pathological progression of synucleinopathies. I believe that the manuscript meets the journal’s high standard for publication. I am looking forward to seeing more papers written by the same authors.

Thank you.

Best regards, 

Reviewer